# Biphasic α2β1 Integrin Expression in Breast Cancer Metastasis to Bone

**DOI:** 10.3390/ijms22136906

**Published:** 2021-06-27

**Authors:** Milene N.O. Moritz, Alyssa R. Merkel, Ean G. Feldman, Heloisa S. Selistre-de-Araujo, Julie A. Rhoades (Sterling)

**Affiliations:** 1Program in Evolutionary Genetics and Molecular Biology, Federal University of Sao Carlos, Sao Carlos, SP 13565-905, Brazil; milenemoritz@gmail.com (M.N.O.M.); hsaraujo@ufscar.br (H.S.S.-d.-A.); 2Division of Clinical Pharmacology, Department of Medicine, Vanderbilt University Medical Center, Nashville, TN 37232, USA; alyssa.r.merkel@vumc.org; 3Center for Bone Biology, Vanderbilt University Medical Center, Nashville, TN 37232, USA; 4Vanderbilt Graduate School Program in Biomedical Sciences, Vanderbilt University Medical Center, Nashville, TN 37232, USA; egfeldman12@gmail.com; 5Department of Physiological Sciences, Federal University of Sao Carlos, Sao Carlos, SP 13565-905, Brazil; 6Department of Cancer Biology, Vanderbilt University Medical Center, Nashville, TN 37232, USA; 7Veterans’ Affairs Tennessee Valley Healthcare System, Nashville, TN 37232, USA

**Keywords:** tumor-induced bone disease, breast cancer, bone metastasis, integrin α2β1

## Abstract

Integrins participate in the pathogenesis and progression of tumors at many stages during the metastatic cascade. However, current evidence for the role of integrins in breast cancer progression is contradictory and seems to be dependent on tumor stage, differentiation status, and microenvironmental influences. While some studies suggest that loss of α2β1 enhances cancer metastasis, other studies suggest that this integrin is pro-tumorigenic. However, few studies have looked at α2β1 in the context of bone metastasis. In this study, we aimed to understand the role of α2β1 integrin in breast cancer metastasis to bone. To address this, we utilized in vivo models of breast cancer metastasis to bone using MDA-MB-231 cells transfected with an α2 expression plasmid (MDA-OEα2). MDA cells overexpressing the α2 integrin subunit had increased primary tumor growth and dissemination to bone but had no change in tumor establishment and bone destruction. Further in vitro analysis revealed that tumors in the bone have decreased α2β1 expression and increased osteolytic signaling compared to primary tumors. Taken together, these data suggest an inverse correlation between α2β1 expression and bone-metastatic potential. Inhibiting α2β1 expression may be beneficial to limit the expansion of primary tumors but could be harmful once tumors have established in bone.

## 1. Introduction

Breast cancer is the most common cancer among women and the second leading cause of cancer-related deaths [1]. Despite advances in early detection and therapeutic options for patients, metastatic disease remains the leading cause of patient mortality. Metastatic breast cancer cells have a high preference for bone. It is estimated that 70% of patients with metastatic disease will have bone involvement [2,3], resulting in increased fracture risk, hypercalcemia, increased morbidity and decreased quality of life [4]. Despite the high prevalence of bone metastases, the pathology and risk factors of breast cancer metastasis to bone are not fully understood.

Recent studies have revealed that the expression profile of primary tumors and composition of the surrounding extracellular matrix (ECM) are important factors contributing to tumor progression and metastasis [5,6,7]. Specifically, the expression of cell surface adhesion receptors, such as integrins, have been shown to be prognostic [8,9,10]. Integrins are αβ heterodimeric transmembrane receptors that support cell adhesion to the extracellular matrix (ECM) and trigger intracellular signaling that can modify cellular behavior [11,12]. Alterations in integrin expression are commonly found in cancer and have been linked to increased tumor proliferation, invasion, and secondary site colonization, as well as decreased patient survival [13,14].

α2β1 integrin has been implicated as an important target in cancer progression due to its critical role in a variety of cancers [15]. Studies have shown that α2β1 integrin is a marker of malignant progression in prostate cancer [16,17,18], liver cancer [19,20], gastric cancer [21,22,23], and melanoma [24]. However, in breast cancer, there is conflicting evidence for the role of α2β1 integrin. While some studies suggest that the loss of α2β1 promotes breast cancer metastasis [25,26], other studies suggest that high α2β1 expression correlates with a metastatic phenotype [19,27,28]. It is believed that α2β1 integrin may also play an important role in bone metastasis due to its high affinity to bind collagen type 1 (which is the main component of the organic part of bone) [29]. While α2β1 has been shown to promote skeletal metastases in prostate cancer [18,30], very few studies have looked at this integrin in the context of breast cancer metastasis to bone.

Thus, this study aimed to investigate the role of α2β1 integrin in breast cancer metastasis to bone. We hypothesized that α2β1 integrin promotes a bone-metastatic potential of breast cancer cells. To test this hypothesis, we used in vivo mouse models of cancer metastasis to bone (orthotopic: mammary fat pad—MFP, metastasis and colonization: intracardiac—IC, and establishment in bone: intratibial—IT). In order to investigate tumor progression and metastasis with respect to α2β1 expression, we developed high expressing MDA-MB-231 breast cancer cells by transfecting cells with an α2 DNA plasmid (OE-α2). In this study, we demonstrate that α2β1 integrin promotes tumor development at the primary site and metastasis to bone but has no effect on bone destruction once tumors have established in bone.

## 2. Results

### 2.1. α2β1 Expression Correlates with an Invasive and Migratory Phenotype

Integrin α2β1 expression is often upregulated in metastatic cancer cells [8,31,32]. Whole exome sequencing of tumor biopsies from patients collected under the Metastatic Breast Cancer Project revealed that metastatic primary tumors have higher ITGA2 and ITGB1 copy number compared to non-metastatic primary tumors (Figure 1A). In order to study the effect of elevated α2β1 expression on breast tumor behavior, we generated a model of MDA-MB-231 breast cancer cells with high α2β1 by stably transfecting a bone-derived clone of MDA-MB-231 (MDA-Bone) with an expression plasmid for α2 (MDA-OEα2) or an empty vector control (MDA-Ctrl). Manipulation of integrin expression and signaling was confirmed by qPCR and western blot analysis (Figure 1B,C). Although we only introduced an α2 expression plasmid into the cells, we were able to achieve significantly higher mRNA and protein expression for both α2 and β1 subunits compared to Ctrl cells. Downstream integrin signaling was also shown to be activated in MDA-OEα2 cells (Figure 1C).

We further wanted to analyze these cells for changes in proliferation and invasive or migratory phenotype in response to enhanced integrin expression. While there was no change in tumor proliferation, we found that tumor cells with high α2β1 expression were more invasive and migratory (Figure 1D,F). A higher number of MDA-OEα2 cells migrated in a transwell invasion assay compared to MDA-Ctrl cells. Furthermore, a scratch assay revealed an increased migration rate in MDA-OEα2 cells.

### 2.2. α2β1 Integrin Promotes Primary Tumor Growth and Dissemination to Bone

Current evidence suggests that α2β1 integrin can act as both a tumor suppressor [25,26,33] and a tumor promoter [19,27,28] in breast cancer and seems to be dependent on tumor status [34]. While most of these studies have looked at invasion and dissemination to soft tissue sites, few studies have elucidated the role of α2β1 integrin in breast cancer dissemination to the bone. Here, we used an in vivo mammary fat pad model of human breast cancer to investigate the effect of elevated α2β1 expression on primary tumor growth, flow cytometry analysis to determine changes in circulating tumor cells (CTCs) and dissemination to bone, and histology analysis to investigate metastases to the lung or bone.

Tumor growth analysis revealed that breast cancer cells expressing high levels of α2β1 have increased growth in vivo (Figure 2A) and larger tumors at sacrifice (Figure 2B,C) compared to control tumors. Immunohistochemical analysis for the α2 integrin subunit confirmed higher expression in MDA-OEα2 tumors. Interestingly, we found that α2 expression in the control tumors was higher at the periphery of the tumor, while α2 overexpressing tumors had high expression throughout the tumor (Appendix A). These data support our hypothesis that α2β1 is needed for migration and invasion from the primary site.

Using a novel flow cytometry technique for detecting disseminated tumor cells in models of low tumor burden using the human cell marker CD298 [35], we analyzed plasma for the presence of CTCs and bone marrow for the presence of disseminated tumor cells (DTCs) (gating scheme can be found in Appendix A). Consistent with the tumor growth analysis, we found that mice injected with MDA-OEα2 cells had an increase in the number of disseminated tumor cells in the bone marrow compared to mice injected with MDA-Ctrl cells (Figure 2D). The presence of tumor cells in the bone marrow was confirmed by histomorphometry analysis with H&E staining (Appendix A). Although not statistically significant, there was a trending increase in the presence of CTCs for mice given MDA-OEα2 cells (Figure 2E). While there was an observed increase in bone metastases (Appendix A), histological analysis revealed no significant difference in lung metastases (Figure 2F). Taken together, these data reveal that high α2 expression in tumors at the primary site results in increased tumor growth and increased dissemination to the bone. Higher α2 expression at the periphery of the tumor and the presence of CTCs also suggest that α2β1 integrin may be creating a more invasive and metastatic phenotype in these breast cancer cells.

### 2.3. Breast Cancer Cells with High α2β1 Expression Have Increased Colonization in the Bone, but Have No Effect on Bone Destruction

Integrins have been shown to play an important role at many stages of the metastatic cascade [13]. Specifically, β1 integrins have been implicated in extravasation from the vasculature and colonization into secondary sites [13,32,36]. To study the role of α2β1 integrin in tumor cell colonization of the bone, we used an in vivo metastasis model where tumor cells are introduced directly into the vasculature via intracardiac (IC) injection. Four–six-week-old female athymic nude mice were injected with MDA-OEα2 or -Ctrl cells. Histological analysis revealed that high expression of α2β1 integrin on the surface of tumor cells increased the amount of tumor cells that colonized the bone but had no effect on subsequent bone destruction (Figure 3A–C). There was significantly higher % tumor area in the tibias of mice given MDA-OEα2 cells compared to mice given MDA-Ctrl cells, but no significant differences were found in bone volume (%BV/TV) by μCT or lesion area by X-ray. These data support our findings in the MFP model that α2β1 expression correlates with an increase in breast tumor dissemination to bone.

To study the effect of α2β1 expression on tumors that have already established in bone, we used an in vivo model of tumor growth in bone. MDA-OEα2 or MDA-Ctrl cells were injected in the right tibia (IT injection) of 4–6-week-old athymic nude mice. PBS was injected into the contralateral limb for a non-tumor control. Interestingly, high α2 expression in established bone metastases had no effect on overall tumor burden and bone destruction (Figure 3D–F). Substantial bone destruction was observed by X-ray and μCT analysis for mice given MDA-Ctrl cells and for mice given MDA-OEα2 cells. H&E staining showed significant tumor burden in the tibias of both sets of mice, with no significant difference in % tumor area between OEα2 and Ctrl cells.

### 2.4. Osteolytic Breast Tumor Cells Have Decreased α2β1 Expression

The in vivo data reveals α2β1 to be a tumor promoter at earlier stages of metastasis, such as invasion and extravasation, but seem to have no effect on tumors already established in bone. To further understand the phenotype, we wanted to evaluate differences in gene expression for tumors that metastasize to bone and cause bone destruction versus primary tumors. We analyzed the mRNA and protein expression profiles of our bone-metastatic clone of MDA-MB-231 cells (MDA-Bone) and the parental MDA-MB-231 cells from ATCC (MDA-Parental) and found that bone-metastatic cells have decreased integrin signaling (Figure 4A,B). MDA-Bone cells have decreased expression of α2 and β1 subunits and decreased protein expression of the downstream signaling factors SRC, RhoGTP, and ROCK. Due to its critical role in bone metastases [37], the integrin subunit β3 was also evaluated; however, there was no significant difference in β3 mRNA or protein expression, suggesting that these changes in integrin signaling are driven primarily by α2β1.

This decrease in α2β1 integrin expression in bone metastases was also observed in vivo. Immunohistochemical analysis revealed that tumors in the bone have significantly lower expression of α2 compared to tumors in the mammary fat pad (Figure 4C). Publicly available genome expression datasets of metastatic breast cancer patients were analyzed to confirm the clinical relevance of our findings. Single-cell microarray analysis of circulating tumor cells (CTCs) isolated from peripheral blood and disseminated tumor cells (DTCs) isolated from bone marrow aspirates of breast cancer patients [38] (GEO accession GSE27574) reveals that DTCs have a decrease in copy number of *ITGA2* and *ITGB1* compared to primary breast tumor samples and CTCs (Figure 4D). Whole exome sequencing from the Metastatic Breast Cancer Project [39,40] was analyzed for putative copy number alterations of *ITGA2* and *ITGB1* in biopsies of primary tumors with no evidence of metastatic disease (non-metastatic primary), biopsies of bone metastases, and biopsies of soft tissue metastases (Figure 4E). While no significant difference was observed for *ITGA2*, *ITGB1* was significantly increased in soft tissue metastases, compared to primary tumors, and bone metastases had lower *ITGB1* than soft tissue metastases.

2.5. α2β1 Integrin Expression Is Inversely Correlated with Osteolytic Gene Expression

It is well documented that once tumors metastasize to bone, they can respond to stimuli from the bone microenvironment to adapt a bone-destructive phenotype [41,42]. Once in the bone, breast tumors begin to secrete parathyroid hormone-related protein (PTHrP) to stimulate osteoclastogenesis and bone destruction [43,44,45]. This increased bone destruction causes the release of matrix-derived proteins such as transformation growth factor β (TGF-β), which then feeds back on the tumor cells to promote further production of PTHrP [46,47], which is regulated by the transcription factor Gli2 [48,49]. To evaluate the expression patterns of genes involved in tumor-induced osteolysis with respect to α2β1 integrin, we performed qPCR and western blot analysis for PTHrP and Gli2 (Figure 5A,B) and TGFβrII and RUNX2 (Appendix A) in MDA-Ctrl or MDA-OEα2 cells, and MDA-Parental or MDA-Bone cells. MDA-MB-231 cells overexpressing α2 integrin had decreased PTHrP and Gli2 expression compared to bone-metastatic cells and Ctrl cells, while no significant change was observed for TGFβrII and RUNX2. Comprehensive RNA sequencing data collected as a part of the MET500 cohort [50] was analyzed for gene expression of *Gli2*, *PTHLH*, *ITGB1*, and *ITGA2* in metastatic breast cancer samples. Spearman correlation analysis of gene signatures in metastatic biopsies of breast cancer reveal a significant (*p* < 0.001) negative correlation between *PTHLH* and *ITGA2* (*p* < 0.001), *PTHLH* and *ITGB1* (*p* < 0.01), *Gli2* and *ITGA2* (*p* < 0.001), and *Gli2* and *ITGB1* (*p* < 0.0001) (Figure 5C,D).

Taken together, these data support the hypothesis that once tumors metastasize to the bone microenvironment, they undergo genetic changes and adapt a bone destructive phenotype. While expression of α2β1 integrin plays an important role in tumor invasion, extravasation, and dissemination, once tumors establish in bone, they turn off the expression of α2β1 and turn on expression of genes important for growth and survival in bone and bone destruction.

## 3. Discussion

Here, we present a novel dual role for α2β1 integrin in breast cancer metastasis to bone. In the primary site, tumors with increased α2β1 expression are larger in size and have increased dissemination to and colonization of bone. However, once tumors establish in bone and a secondary metastasis is formed, those metastatic tumors turn off α2β1 signaling and adopt a bone-destructive phenotype. Consistent with our results, a recent study demonstrated that α2 integrin knockdown decreased tumor cell migration and invasion as well as mammosphere formation, relating α2 integrin to promoting breast cancer metastasis [51].

These data suggest that α2 integrin is creating an oncogenic switch in these tumor cells. When MDA-MB-231 cells were transfected with an α2 overexpression plasmid, expression of the β1 integrin subunit also increased (Figure 1B,C). This increased β1 expression could be a compensatory effect from the α2 overexpression since α2 exclusively dimerizes with β1 to form the α2β1 heterodimer [15]; however, the mechanism of α2-integrin-mediated β1 upregulation is poorly understood and deserves to be researched further. β1 integrin can also dimerize with other α integrins, and several of these integrins such as α5β1, α1β1, and α3β1 have been shown to regulate tumor cell invasion and promote tumor metastasis [52]. It is possible that increased β1 expression by these tumor cells could be contributing to this phenotype through mechanisms other than α2β1, but this is beyond the scope of this manuscript.

We believe α2β1 is promoting tumor metastasis through regulation of EMT and MET. Breast cancer cells often undergo an epithelial–mesenchymal transition (EMT) in order to migrate, invade, and disseminate from the primary site and once a secondary metastasis is formed, often go back through a mesenchymal–epithelial transition (MET) in order to establish and proliferate at the secondary site [53]. α2 and β1 integrins and their downstream signaling factors have been shown to regulate EMT and MET, resulting in alterations of cancer cell behavior [54,55]. During EMT, the kinases FAK and Src control integrin-mediated cell adhesion and migration; in epithelial cancer progression, the migratory capacity and the intercellular contact suppression are enhanced via Src activation and downregulation of FAK [56].

In the present study, we have shown that OE-α2 cells, which presented with an increase in migratory behavior, have low levels of phosphorylated FAK and high levels of phosphorylated Src compared to control cells (Figure 1). This decrease in phosphorylated FAK contributes to attenuated focal adhesions and consequently cell detachment while Src activation suppresses cell–cell adhesion, causing an increased migratory phenotype. Furthermore, migratory behavior of primary tumor cells seems to be related to β1 dephosphorylation. Migration events are precisely regulated via integrin-mediated cell adhesion with coordinated cycles of attachment and detachment from the ECM [57]. While OEα2 cells presented with high levels of total β1 integrin, its activation was not observed, indicating that its binding to the ECM was disabled, contributing to cell detachment.

The current literature about α2β1 integrin in cancer progression has shown dual roles as both a tumor suppressor and a tumor promotor [15,16,26,58]. We hypothesized that this inconsistency might be better understood by observing its function at each step of metastasis. Therefore, in order to better elucidate the role of α2β1 integrin in each step of the breast cancer–bone metastatic cascade, we used three different tumor mouse models to simulate (i) primary tumor growth and invasion (MFP), (ii) tumor bone colonization (IC), and (iii) tumor establishment in bone and bone destruction (IT). Here, we establish that α2β1 integrin acts as a tumor promoter in primary disease, while it has an inverse correlation with a bone destructive phenotype in bone metastases.

We understand that the apparent contradiction is due to the fact that the expression of α2β1 integrin varies along the metastatic cascade. Microarray database analysis collected from NCBI showed that tumor cells isolated from bone marrow have fewer copy numbers of *ITGA2* and *ITGB1* compared to primary tumor cells or circulating tumor cells (Figure 4D). Additionally, analysis of whole exome sequencing of tumor biopsies from the Metastatic Breast Cancer Project showed a decrease in copy number of *ITGB1* and variable expression of *ITGA2* between primary tumors, soft tissue metastases, and bone metastases (Figure 4E). Expression analysis in patient data was corroborated in our cell lines. A bone metastatic clone of the MDA-MB-231 breast cancer cell line (MDA-Bone) had lower levels of α2β1 integrin compared to MDA-Parental cells from primary breast tumor (Figure 4A,B). Given this variable expression between primary and metastatic disease and the variable expression found in circulating tumor cells [13,59], we believe that that the bone microenvironment is having a fundamental influence on the integrin expression in bone metastases.

Tumors in the bone often rely on alternative signaling pathways in order to survive in the bone/bone marrow, making them fundamentally different from tumors at other sites [42]. Here, we observed that the lack of α2β1 integrin in bone metastases is associated with a decrease in phosphorylated Src, an increase in phosphorylated FAK, and decreased amounts of RhoA and ROCK1 (Figure 4), indicating that these cells, once established in bone, lose their migratory phenotype. Rho-ROCK signaling is involved with migration, controlling the organization of actin cytoskeleton and cell motility by activating a number of downstream targets [60], while this increased FAK expression and decreased Src expression contributes to the tumor cells being able to bind to the bone ECM and establish in the bone microenvironment. Furthermore, the decrease in α2β1 integrin expression correlated with an increase in the genes *PTHLH* and the hedgehog transcription factor *Gli2* (Figure 5), which are involved in tumor-induced osteolysis [61,62].

Interestingly, bone metastatic tumor cells have decreased β1 expression while expressing high β3 integrin levels, providing evidence that integrin switching is occurring in the bone microenvironment. It has been reported that inactivation of β1 integrins cause β3 integrin switching and TGF-β induced breast cancer progression [32,63]. β3 integrin has been associated with breast cancer bone metastasis and its expression is increased in bone metastatic tumors [64]. Our group has previously reported that under mechanical force, αvβ3 integrin interacts with TGF-β rII, causing the tumor cells to switch to a bone destructive phenotype [65].

Considering the data presented in this study, we suggest that α2β1 integrin expression may be involved in the oncogenic switch of primary tumors to a migratory and metastatic phenotype, while inactivation of α2β1 by bone microenvironmental influences, such as TGF-β and mechanical force, promote tumor-induced bone disease. Inhibiting α2β1 expression may be beneficial to limit the expansion of primary tumors but could have detrimental effects to TGF-β-mediated tumor effects or tumors already established in bone. The paradigm of how integrins, ECM and TGF-β functions are related is still unclear [66], and therefore, it is necessary to further investigate this integrin-mediated signaling to better understand how to prevent breast cancer metastasis to bone.

## 4. Materials and Methods

### 4.1. Cell Culture

Human MDA-MB-231 breast cancer adenocarcinoma cells were obtained from American Type Tissue Culture Collection (ATCC), and bone-derived cells were generated by our laboratory [67]. Cell lines were maintained in Dulbecco’s Modified of Eagle’s Medium (DMEM, Corning, Manassas, VA, USA) and cultured in a humidified incubator at 37 °C under 5% CO_2_. All media was supplemented with 10% fetal bovine serum (FBS, Peak Serum, Wellington, CO, USA) and 1% Penicillin and Streptomycin (Mediatech, Manassas, VA, USA). 3.6 × 10^5^ cells were plated in 6-well plates for downstream experiments.

### 4.2. Transfection

To generate MDA-OEα2 cells, bone-derived MDA-MB-231 (MDA-bone) cells were transfected with an α2 expression DNA plasmid. A total of 3.6 × 10^5^ cells were plated in a 6-well plate and cultured for 24 h before transfection. MDA-bone cells were stably transfected with 5 µg of m-Emerald-Integrin-Alpha2-N-18 plasmid (# 54128, Addgene, Watertown, MA, USA) (MDA-OEα2) or 5 µg of p-Dest m-Cherry N1 plasmid control (# 31907, Addgene, Watertown, MA, USA) (MDA-Ctrl). All transfections were performed using Lipofectamine LTX with plus reagent (# 15338030, Invitrogen, Carlsbad, CA, USA) per manufacturer’s instructions. Cells were selected in medium containing 700 µg/mL of geneticin (G418, Corning, Manassas, VA, USA). Colonies were isolated, expanded and maintained in a medium with 500 µg/mL G418.

### 4.3. Quantitative Real-Time PCR

To measure gene expression changes, cells were harvested with trypsin and total RNA was extracted using the RNeasy Mini Kit (Qiagen, Hilden, Germany). mRNA reverse transcription was performed using 1 μg total RNA and the qScript cDNA supermix, (Quanta Biosciences, Beverly, MA, USA) following manufacturer’s instructions. The expression of ITGA2 (Hs04332845), ITGB1 (Hs00559595), ITGB3 (Hs01001469), PTHrP (Hs00174969), and Gli2 (Hs01119974) was measured in triplicate by qRT-PCR using validated TaqMan primers from Applied Biosciences (Carlsbad, CA, USA) with the 7500 Real-Time PCR System (Applied Biosystems, Foster City, CA, USA) using the following cycling conditions: 95 °C for 15 s and 60 °C for 1 min, preceded by an initial incubation period of 95 °C for 10 min. Quantification was performed using the absolute quantification for human cells method using 18S or GAPDH as an internal control.

### 4.4. Western Blot Analysis

To measure protein expression changes, cells were harvested in RIPA buffer containing a cocktail of proteases and phosphatase inhibitors (Thermo Scientific, Rockford, IL, USA). Equal protein amount (20 µg) was prepared and run on a 4–20% SDS-PAGE gel. Proteins were transferred to a nitrocellulose membrane and blocked with 5% BSA for 1 h at room temperature, followed by incubation with primary antibodies (Table 1) overnight at 4 °C or GAPDH (1:5000 Cell Signaling Technology # 2118) loading control. Membranes were then incubated with a secondary antibody at 1:5000 (anti-mouse, Santa Cruz # sc-2005 or anti-rabbit, Santa Cruz # sc-2004) and bands were detected by chemiluminescence using a Chemidoc Touch gel imager (Bio-Rad, Hercules , CA, USA).

### 4.5. Proliferation Assay

MDA-Ctrl or MDA-OEα2 cells were plated in quadruplet in a 96-well plate at a density of 1 × 10^4^ cells/well. Cell proliferation was measured at 0, 24, 72, or 120 h using the CellTiter 96 Aqueous Non-Radioactive Cell Proliferation Assay kit (Promega, Madison, WI, USA) per the manufacturer’s instructions. Absorbance values were measured on a Synergy 2 microplate reader (Biotek, Winooski, VT, USA) at an optical density (OD) of 450 nm.

### 4.6. Invasion Assay

MDA-Ctrl or MDA-OEα2 cells (2.0 × 10^5^ cells) were plated in the top of 24 mm transwells with an 8.0 µm pore size (Corning, # 3428). Complete media or media + 40 μg/mL Collagen I (Gibco # A10483-01) was used in the bottom of the 6-well plate as a chemoattractant. For negative controls, serum-free media was used. After 24 h, non-migrated cells were removed from the inside of inserts with cotton swabs and cells that migrated to the bottom of the transwell were fixed with 5% glutaraldehyde and stained with 1% crystal violet in 2% ethanol. Transwells were imaged at 10× magnification using an Olympus CKX41 inverted microscope. Three representative images were taken per transwell, and the average number of migrated cells was calculated using ImageJ analysis.

### 4.7. Scratch Assay

MDA-Ctrl or MDA-OEα2 cells were plated into 6-well culture plates and incubated with 10% FBS media to reach confluence. The monolayers were carefully wounded using a 1000-μL pipette tip, and cellular debris were removed by PBS wash. The wounded monolayers were cultured for 48 h to monitor wound healing. Images at 10× were taken at times 0, 2, 4, 6, 12, 24, and 48 h post wounding using an Olympus CKX41 inverted microscope. Wound width was calculated by ImageJ analysis, and migration rate was calculated as (initial wound width—wound width at 48 h)/48 h.

### 4.8. In Vivo Studies

All animal experiments were carried out in compliance with the Vanderbilt University Institutional Animal Care and Use Committee (protocol # M1600176) and the National Institute of Health guidelines. Four–six-week-old female athymic nude mice (Envigo, Indianapolis, IN, USA) were used for all animal experiments. To investigate primary tumor growth, 5 × 10^5^ tumor cells were injected into the 4th inguinal mammary fat pad (MFP model, *n* = 8). To investigate tumor extravasation and colonization to bone, 1 × 10^5^ tumor cells were injected into the left cardiac ventricle (intracardiac, IC model, *n* = 12). To investigate tumor growth at the metastatic site, 1 × 10^5^ tumor cells were injected into the right tibia and PBS into the left tibia as control (intratibial, IT model, *n* = 8). Mice were injected with either MDA-Ctrl or MDA-OEα2 tumor cells. Mice were sacrificed after 3–4 weeks, and primary tumor, hindlimbs, blood, spleen, lung, and liver were collected for ex vivo analysis.

### 4.9. Quantification of Tumor Growth

Primary tumor size was monitored by 3x weekly caliper measurements. Tumor height and width were measured in vivo with calipers, and tumor volume was calculated using the formula V = (W^2^ × L)/2. Tumor size was also analyzed at sacrifice. Weight was measured on an analytical balance, and tumor volume (length × width × height) was measured with calipers after the tumor was excised from the mouse.

### 4.10. X-ray Analysis

Bone destruction was monitored by weekly X-ray imaging on a XR-60 Faxitron digital radiography system (Hologic, Marlborough, MA, USA) at 35 kVp for 8 sec, beginning at 1-week post tumor inoculation and continuing for the remainder of the study. Osteolytic lesions were quantified bilaterally in the humeri, femora, and tibiae using ROI analysis on the Metamorph imaging analysis software (Molecular Devices, San Jose, CA, USA).

### 4.11. Micro Computed Tomography

For trabecular bone volume analysis, tibiae were scanned on a Scanco μCT40 (Scanco Medical, Wangen-Brüttisellen, Switzerland) at 70 kVp with a 12 µm voxel size and an integration time of 300 ms. Scans were contoured to analyze trabeculae in the metaphasis region starting 10 slices below the growth plate and continuing 100 slices in the distal direction. Images were reconstructed using the Scanco Medical Imaging software and calculations for bone volume fraction (BV/TV, bone volume over total volume) were calculated by the software.

### 4.12. Histomorphometry

Excised tissues were fixed in formalin for 48 h, and then stored at 4 °C in 70% ethanol. Bones were decalcified in 20% EDTA at 4 °C for up to 2 weeks. Tissues were processed, embedded in paraffin, and 5 μm thick serial sections were cut on a microtome (Leica Biosystems, Wetzlar, Germany). Sections were stained with Hematoxylin and Eosin (H&E) to analyze % tumor area. Tumor burden was analyzed in the hindlimbs by freehand ROI analysis in Metamorph and was measured as a percentage of total bone marrow area.

Immunohistochemistry (IHC) staining was performed on primary tumor and hindlimb sections to evaluate the expression of α_2_ integrin (1:250, Abcam # ab13355). Briefly, antigen retrieval was performed with citrate buffer followed by blocking in 5% goat serum for 1 h and primary antibody incubation overnight at 4 °C. Sections were incubated in an anti-rabbit secondary antibody (Santa Cruz, # sc2004) for 2 h at room temperature and detected with the NovaRed peroxidase kit (Vector Labs, Burlingame, CA, USA). The % stained area over the total area was quantified using Metamorph imaging analysis software.

### 4.13. Flow Cytometry

Blood was collected from mice via cardiac puncture immediately prior to sacrifice. Plasma was isolated and prepped for flow cytometry to analyze circulating tumor cells [68]. Bone marrow was flushed from the femurs of mice by centrifugation. Cells were resuspended in red blood cell lysis buffer, incubated for 5 min on ice, spun down, and washed twice with PBS. Up to 1 × 10^6^ plasma or bone marrow cells were stained with 175ng CD298 antibody (BioLegend, Cat # 341704) for 30min on ice in the dark [35]. Samples were run on a 5-laser BD-LSRII in the Vanderbilt Medical Center Flow Cytometry Shared Resource core. % CD298^+^ cells in the bone marrow or blood were analyzed using FlowJo Software (BD, version 10.7.1, Franklin Lakes, NJ, USA).

### 4.14. Patient Data Analysis

Publicly available genome datasets of metastatic breast cancer patients were accessed and analyzed for gene expression patterns of *ITGA2*, *ITGB1*, *PTHLH* and *Gli2*. Single-cell microarray data from the study “High-resolution analysis of copy number changes in circulating and disseminated tumor cells in breast cancer patients” [38] was obtained from the NCBI gene expression omnibus [69] (GEO accession GSE27574). Copy number data for *ITGA2* (NM_002203) and *ITGB1* (NM_133376) were extracted from the dataset. One-way ANOVA was performed to assess differences in gene copy number between bone marrow disseminated tumor cells (DTCs, N = 24), peripheral blood circulating tumor cells (CTCs, N = 28) and primary tumor samples (N = 3).

Whole genome sequencing data for biopsies of metastatic breast cancer patients were collected from The Metastatic Breast Cancer Project [39,40] database through CBioPortal [70,71,72]. Putative copy number alterations were analyzed for *ITGA2* and *ITGB1*. Samples were selected based on PATH procedure location (primary = breast; bone metastases = bone, chest wall, and bone marrow; soft tissue metastases = liver, brain, lung, and soft tissue). Samples from patients who had already received therapy before collection were excluded in order to assess baseline gene expression. Primary tumors were further subdivided into non-metastatic (patients with no metastatic disease present at the time of sample collection and no metastatic disease at follow up) and metastatic (patients with metastatic disease present at the time of sample collection). One-way ANOVA was performed to analyze differences in gene copy number between non-metastatic primary tumors (N = 14), metastatic primary tumors (N = 42), bone metastases (N = 8), and soft tissue metastases (N = 10).

RNA sequencing data for *ITGA2* (ENSG00000164171.10), *ITGB1* (ENSG00000150093.18), *PTHLH* (ENSG00000087494.15) and *Gli2* (ENSG00000074047.20) were collected from the MET500 cohort [50] using the UCSC Xena browser [73]. Samples were selected based on the primary site (breast, N = 120). Spearman correlation analysis was performed on *Gli2* vs. *ITGA2*, *Gli2* vs. *ITGB1*, *PTHLH* vs. *ITGA2*, and *PTHLH* vs. *ITGB1*. RNA expression.

### 4.15. Statistical Analysis

Statistical analyses were performed using one-way ANOVA for multiple comparisons and Mann–Whitney or two-tailed Student’s *t*-tests. All statistical analyses and graphs were generated using GraphPad Prism software. Data values are presented as mean ± SEM, and *p* < 0.05 was considered statistically significant. * *p* < 0.05, ** *p* < 0.01, *** *p* < 0.001, **** *p* < 0.0001.

## 5. Conclusions

This study elucidates the role of α2β1 integrin in different stages of breast cancer metastasis to bone and highlights the biphasic expression pattern of this integrin. High α2β1 expression in tumors at the primary site drives tumor growth, invasion, and dissemination while loss of α2β1expression in tumors in the bone microenvironment drives tumor growth and bone destruction (Figure 6). This work provides suggestive preclinical evidence in mice that α2β1 could be used as a biomarker for bone metastatic potential of breast cancer. Therapeutic targeting of α2β1 integrin could be beneficial for primary disease or early-stage breast cancer patients without worsening disease at other sites or damaging the bone or bone marrow.

## Figures and Tables

**Figure 1 ijms-22-06906-f001:**
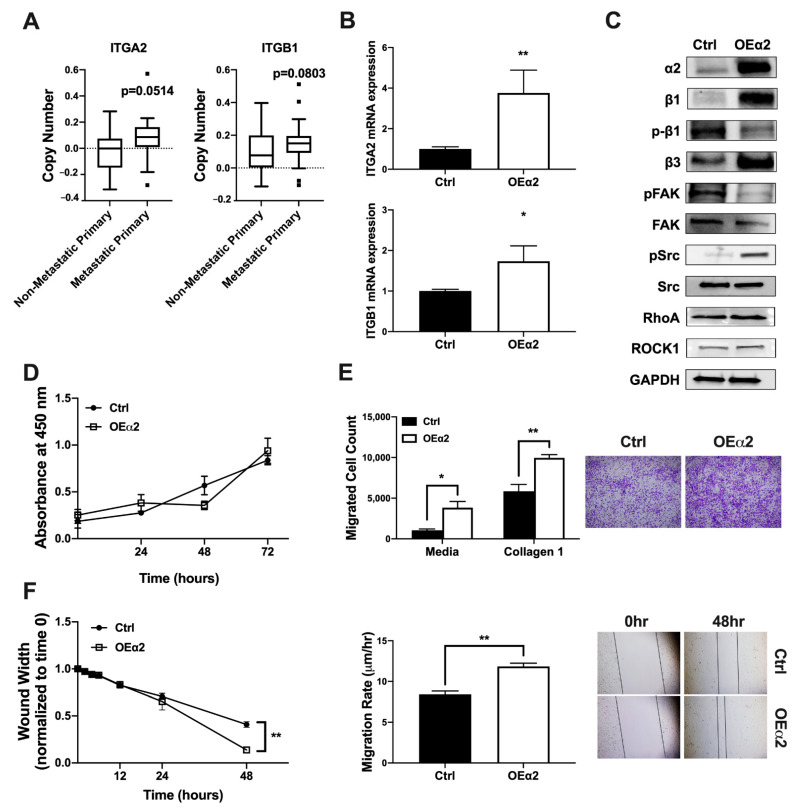
(**A**) Whole exome sequencing of tumor biopsies from patients collected under the Metastatic Breast Cancer Project was analyzed for copy number alterations in ITGA2 and ITGB1. N = 14 non-metastatic primary, N = 42 metastatic primary tumors. Mann–Whitney test. (**B**) qPCR and (**C**) western blot analysis confirmed that cells overexpressing α2 (OEα2) had increased *ITGA2* and *ITGB1* mRNA expression as well as increased α2 and β1 subunit protein expression and activated integrin signaling. (**D**) An MTS proliferation assay showed no significant difference in cell growth at 24, 72, or 120 h in cells expressing high α2 compared to control. (**E**) A significantly higher number of MDA-OEα2 tumor cells compared to MDA-Ctrl cells migrated in a Transwell Invasion Assay using either complete media or media + Collagen 1 as a chemoattractant. (**F**) MDA-OEα2 cells migrated at a faster rate compared to MDA-Ctrl cells in a scratch assay as measured by changes in wound width over time. N = 3 biological replicates. Data presented as fold change over control (Ctrl). Student’s *t*-test * *p* < 0.05, ** *p* < 0.01.

**Figure 2 ijms-22-06906-f002:**
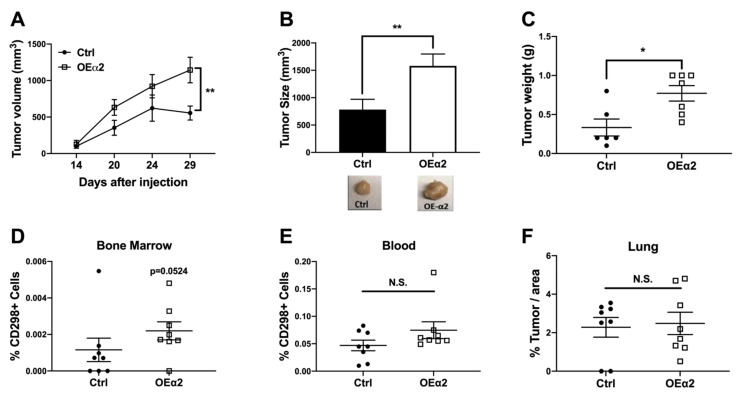
Four–six-week-old athymic nude mice were injected with 5 × 10^5^ MDA-Ctrl or MDA-OEα2 cells into the mammary fat pad. Mice injected with MDA cells overexpressing α2 had increased tumor growth (**A**), larger tumors at sacrifice (**B**), and increased tumor weight (**C**) compared to mice injected with control MDAs. (**D**,**E**) Flow cytometry analysis was performed on bone marrow and blood using a novel protocol to detect the presence of human tumor cells using the marker CD298 (ATP1B3). Mice injected with cells overexpressing the integrin α2 subunit had an increased % CD298^+^ cells in the bone marrow (**D**), and a trending (n.s.) increase in the blood (**E**) compared to mice injected with control cells. (**F**) Lung metastases were quantified by histological analysis, and no difference was observed between OEα2 and Ctrl cells. N = 8 per group. Mice were sacrificed 30 days post tumor inoculation. Two-way ANOVA and Mann–Whitney test. * *p* < 0.05, ** *p* < 0.01.

**Figure 3 ijms-22-06906-f003:**
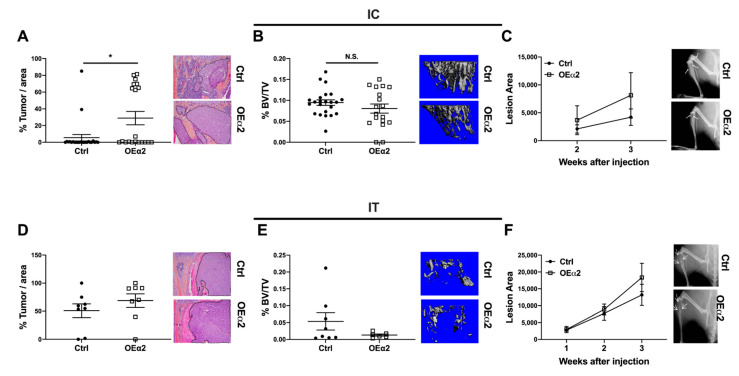
(**A**–**C**) Four–six-week-old athymic nude mice were injected via intracardiac (IC) injection with 1 × 10^5^ MDA-Ctrl or MDA-OEα2 cells. (**A**) H&E staining revealed increased percentage of tumor cells in the tibias of mice injected with MDA-OEα2 cells compared to mice injected with MDA-Ctrl cells. (**B**) μCT analysis and (**C**) X-ray analysis show no change in bone volume and lesion area. N = 12 mice per group, 2 bones analyzed per mouse. IC mice were sacrificed 30 days post tumor inoculation. Mann–Whitney test. * *p* < 0.05. (**D**–**F**) Four–six-week-old athymic nude mice were injected via intratibial (IT) injection with 1 × 10^5^ MDA-Ctrl or MDA-OEα2cells. (**D**) Histomorphometry reveals no difference in tumor area between MDA-OEα2 and MDA-Ctrl injected mice. (**E**) μCT analysis shows no difference in bone volume, and (**F**) X-ray analysis shows no difference in the lesion area. N = 8 per group. IT mice were sacrificed 21 days post tumor inoculation. Mann–Whitney test.

**Figure 4 ijms-22-06906-f004:**
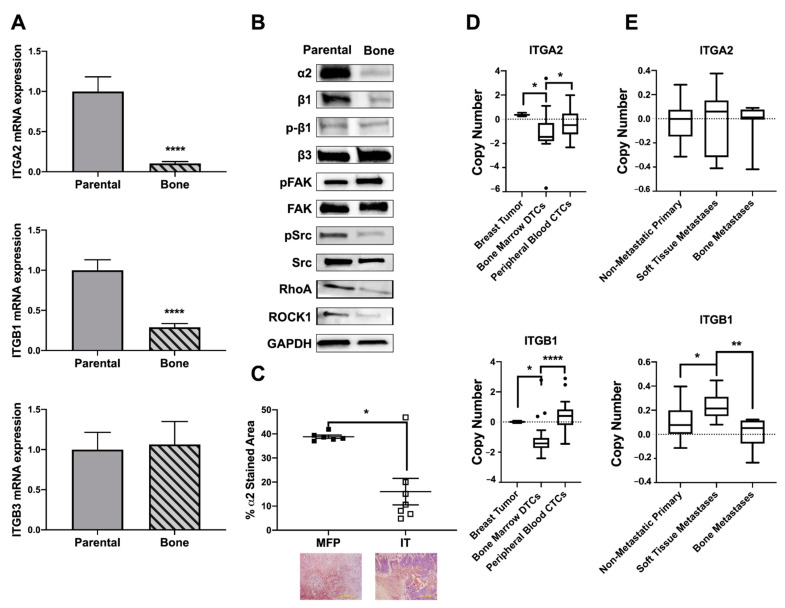
(**A**,**B**) A bone metastatic clone of MDA-MB-231 (Bone) and the parental MDA-MB-231 cells (Parental) were analyzed for integrin expression by (**A**) qPCR and (**B**) western blot analysis. Bone metastatic cells have less expression of the integrin subunits α2 and β1 and downstream integrin signaling compared to parental cells. Data presented as fold change over parental. N = 3 biological replicates. Student’s *t*-test. **** *p* < 0.0001. (**C**) α2 expression was analyzed in vivo by immunohistochemistry revealing that tumors in the bone (IT, intratibial injection) have less tumor expression of α2 compared to tumors in the primary site (MFP, mammary fat pad injection). N = 8 mice per group. Mann–Whitney test. * *p* < 0.05. (**D**) Microarray database analysis collected from the NCBI gene expression omnibus GEO accession GSE27574 revealed that disseminated tumor cells (DTCs) in the bone marrow have fewer copy numbers of *ITGA2* and *ITGB1* compared to primary breast tumors and circulating tumor cells (CTCs). N = 3 breast tumors, N = 24 DTCs, N = 28 CTCs. Kruskal–Wallis test. * *p* < 0.05, **** *p* < 0.0001. (**E**) Whole exome sequencing of tumor biopsies from patients collected under the Metastatic Breast Cancer Project was analyzed for copy number alterations in *ITGA2* and *ITGB1*. Soft tissue biopsies had higher *ITGB1* than non-metastatic primary tumors and tumor biopsies from bone metastases had fewer copy numbers of *ITGB1* compared to soft tissue metastases. No significant difference was observed for *ITGA2.* N = 14 non-metastatic primary, N = 42 metastatic primary tumors, N = 8 bone metastases, N = 10 soft tissue metastases. Kruskal–Wallis test. * *p* < 0.05, ** *p* < 0.01.

**Figure 5 ijms-22-06906-f005:**
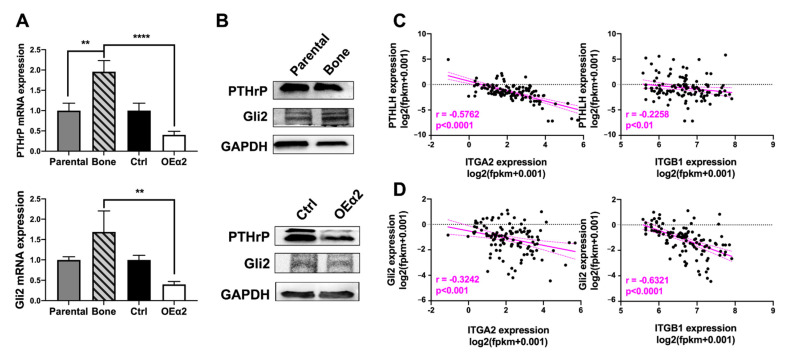
MDA-Parental, MDA-Bone, MDA-Ctrl, and MDA-OEα2 cells were analyzed for osteolytic gene expression by (**A**) qPCR and (**B**) western blot analysis. Tumor cells overexpressing the α2 integrin subunit had decreased PTHrP and Gli2 expression compared to bone and control cells (each set at 1). N = 3 biological replicates. Student’s *t*-test. ** *p* < 0.01, **** *p* < 0.0001. (**C**,**D**) RNA sequencing analysis from metastatic breast cancer biopsies from the MET500 cohort was analyzed for correlation between (**C**) *PTHLH, ITGA1*, and *ITGB1*, and (**D**) *Gli2, ITGA2*, and *ITGB1* gene signatures. Spearman correlation analysis reveal a significant negative correlation between integrin α2β1 and osteolytic genes. N = 120 tumor samples.

**Figure 6 ijms-22-06906-f006:**
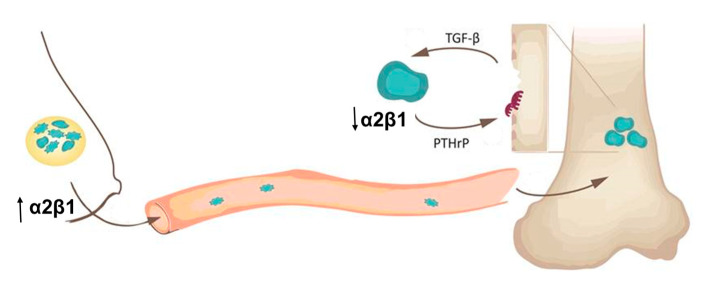
α2β1 integrin expression in primary breast tumors promotes tumor growth, invasion, and dissemination to the bone. However, once established in the bone, α2β1 expression is lost and tumors begin to express genes, such as Gli2 and PTHrP, that promote tumor-induced bone destruction.

**Table 1 ijms-22-06906-t001:** Primary antibodies used for western blot analysis.

	Target	Dilution	Catalog #
**Integrin Receptors**	α2	1:5000	ab133557
	β1	1:5000	sc-8978
	phospho-β1	1:1000	ab5189
	β3	1:1000	sc13579
**Integrin Signaling**	FAK	1:1000	ab40794
	phospho-FAK	1:500	ab81298
	Src	1:1000	2108 ^1^
	phospho-Src	1:1000	2105 ^1^
	RhoA	1:1000	2117 ^1^
	ROCK-1	1:500	sc-5560
**Osteolytic Signaling**	Gli2	1:500	NB600-874
	PTHrP	1:1000	NBP1-59322

^1^ Cell signaling technology.

## Data Availability

“High-resolution analysis of copy number changes in circulating and disseminated tumor cells in breast cancer patients” GSE27574, https://www.ncbi.nlm.nih.gov/geo/query/acc.cgi?acc=GSE27574 (accessed on 7 April 2020). The Metastatic Breast Cancer Project: http://www.cbioportal.org/study/summary?id=brca_mbcproject_wagle_2017 (accessed on 1 April 2020). MET500 Cohort: https://met500.path.med.umich.edu/ (accessed on 17 June 2020).

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
