# Peer review of "Biphasic α2β1 Integrin Expression in Breast Cancer Metastasis to Bone"

_ijms, 2021, doi:10.3390/ijms22136906_

Round 1

Reviewer 1 Report

In this manuscript, authors clearly showed the role of a2b1 integrins in breast cancer metastasis to bone. Contribution of integrin a2b1 was shown not only in acceleration of tumor cell invasion, but also in suppression of bone destruction. From these observations, they argued that the integrin a2b1 would become a fruitful target for preventing tumor bone metastasis.

[Major concerns]

  1. Authors prepared tumor cells with high level a2 integrin expression. However, this cell showed b1 integrin upregulation as well as a2 integrin. All data showed in this paper were prepared using this “a2 integrin upregulation-mediated b1 integrin overexpressed cells”. Therefore, following questions come up to mind.

            ・Mechanisms underlying a2-integrin mediated b1-integrin upregulation ?

            ・Is there possibility that the other b1-integrins, such as a4b1 and a5b1 integrin, plays a key role in breast cancer metastasis than a2b1?  Would you please touch on the other b1integrin heterodimers?

  1. In Figure 1D, overexpression of a2 integrin (with b1 integrin upregulation) showed no effect of tumor cell growth. However, in Figure 2A to C, growth activity of these cells in mice tissue was elevated in 3 assay axes (tumor volume, size, weight).

            ・Would you please explain the reason why these difference occur ?

            ・If tumor growth was accelerated by a2 integrin overexpression , is there possibility that the raised number of CD298+ cells in bone marrow in OEa2 injected mice was carried out by enhanced OEa2 cells growth ?

  1. In Figure 5A, expression of PTHrP and Gli2 in ctrl cell was suppressed to the levels in parental cell, although ctrl cell was prepared by transfecting control vector into MDA-Bone cell, which express both genes in high level. Would you please mention about it ?

Reviewer 2 Report

Majo comments: none

Minor comments: 

Figure 2: please make clear in the figure legend which was the time-point of sacrifice. In methods is written 3-4 weeks. It should be on time-point. "At sacrifice" is too unspecific.

Figure 3: Again, after which time points HE stainings, microCT and x-ray analyses were done? 3 weeks? Please give this information in Fig. legend.

line 225: In the the headline sth. is truncated.
